Extended Abstract Track

# Optimal Latent Transport

**Hrittik Roy**                                 HROY@DTU.DK
**Søren Hauberg**                           SOHAU@DTU.DK
*Section for Cognitive Systems, Technical University of Denmark*

**Editor:** Sophia Sanborn, Christian Shewmake, Simone Azeglio, Arianna Di Bernardo, Nina Miolane

## Abstract

It is common to assume that the latent space of a generative model is a lower-dimensional Euclidean space. We instead endow the latent space with a Riemannian structure. Previous work endows this Riemannian structure by pulling back the Euclidean metric of the observation space or the Fisher-Rao metric on the decoder distributions to the latent space. We instead investigate pulling back the Wasserstein metric tensor on the decoder distributions to the latent space. We develop an efficient realization of this metric, and, through proof of concept experiments, demonstrate that the approach is viable.

**Keywords:** Deep generative models; Wasserstein metric; Earth movers distance; Latent space geometry.

## 1. Introduction

Deep generative models tackle the common problem of inferring a probability distribution from example data. *Generative adversarial networks (GANs)* (Goodfellow et al., 2020) and *variational autoencoders (VAEs)* (Kingma and Welling, 2013; Rezende et al., 2014) both learn a lower-dimensional latent space and a mapping from that to the data space. This latent space $\mathcal{Z}$ describes the data as a lower-dimensional manifold embedded in the data space, $\mathcal{X} \subset \mathbb{R}^D$. In light of this description and since $\mathcal{X}$ is an embedded submanifold, it makes sense to treat $\mathcal{Z}$ as a manifold rather than a Euclidean space (Hauberg, 2018).

The geometry of $\mathcal{Z}$ is determined by the properties of the decoder distributions $\{p_\theta(\cdot|z)|z \in \mathcal{Z}\}$. Early work along this direction focus on the case where the decoder $p_\theta(\cdot|z)$ is Gaussian, and pulls the Euclidean metric from data space into the latent space (Arvanitidis et al., 2018). As classic differential geometry does not lend itself to stochastic models, Eklund and Hauberg (2019) reinterpret the reparametrization trick (Kingma and Welling, 2013) as a random projection, such that Gaussian decoders can be seen as spanning a deterministic manifold in the product space of decoder mean and standard deviation. This then allows us to define a Riemannian metric on $\mathcal{Z}$.

Empirically, this approach has been demonstrated to work well on a range of tasks (Arvanitidis et al., 2018; Kalatzis et al., 2020; Beik-Mohammadi et al., 2021), but it does not generalize beyond Gaussian decoders. In practice, however, Gaussian decoders are rarely used as the associated likelihood is not a good model for e.g. image data. Instead discretized mixtures of logistics and similar likelihood models are used in contemporary VAEs (Salimans et al., 2017; Maaløe et al., 2019).

To construct a more general framework, Arvantidis et al. (2022) recently proposed to pull back an information metric from the space of decoder distributions to the latent space. Specifically, this work pulled the Fisher-Rao metric over decoders into the latent space.

# Extended Abstract Track

While this provides a general approach, the Fisher-Rao metric disregards the original data metric, which often comes with significant physical grounding. For example, the Fisher-Rao metric over distributions of pixel intensities disregards that small intensity fluctuations are less important than large ones.

**In this paper**, we investigate the use of Wasserstein metrics over the decoder distributions. This elegantly combines the observation space metric with the stochasticity of the decoder. We focus on the discretized mixture of logistics likelihood as this is the most popular choice in the VAE literature. We propose an efficient algorithm for evaluating this Wasserstein metric and demonstrate the feasibility of the approach.

## 2. Background

**Variational autoencoders (Kingma and Welling, 2013; Rezende et al., 2014).** In the VAE setting, we have a latent variable model for our data. $z \in \mathbb{R}^d$ denotes the latent variables in a low-dimensional Euclidean space and $x \in \mathcal{X} \subset \mathbb{R}^D$ denotes our observation data which lies on a low-dimensional submanifold of a high-dimensional Euclidean space. The VAE encoder maps points sampled from the dataset to an approximate posterior distribution, denoted by $q_\phi(z|x)$, on the latent space and the decoder maps points sampled from the latent space to a decoder distribution, denoted by $p_\theta(x|z)$, on the data space. We also define a prior on the latent space denoted by $p(z)$. The VAE learns its parameters by maximizing the *evidence lower bound (ELBO)* which is a lower bound of the log-likelihood:

$$\mathcal{L}(\phi, \theta) = \mathbb{E}_{q_\phi(z|x)}[\log p_\theta(x|z)] - \mathrm{KL}(q_\phi(z|x)\|p(z)). \tag{1}$$

**Latent space information geometry (Arvantidis et al., 2022).** The VAE decoder function, $h : \mathcal{Z} \to \mathcal{H}$, maps each point in the sample space to a parameter space $\mathcal{H}$ of a probability distribution $p_\theta(x|z)$ on $\mathcal{X}$. In particular we have $z \mapsto \eta$ and the likelihood is given by $p_\theta(.|\eta)$. The function $h$ that maps latents to parameters allows us to write the likelihood as $p_\theta(x|z)$. For two points $z_1$ and $z_2$ that are arbitrarily close, i.e. $z_2 = z_1 + \epsilon$, we can then define the distance between them to be

$$d^2(z_1, z_2) = \mathrm{KL}(p(\cdot|z_1) \parallel p(\cdot|z_2)). \tag{2}$$

We can define length of curves in the latent space, $\gamma : [0, 1] \to \mathcal{Z}$ to be:

$$l(\gamma) = \lim_{N \to \infty} \sum_{n=1}^{N-1} \sqrt{\mathrm{KL}(p(\cdot|\gamma(n/N)) \parallel p(\cdot|\gamma((n+1)/N)))}. \tag{3}$$

Distances can then be defined as the length of the shortest connecting curves, i.e. $d^2(z_1, z_2) = \inf_\gamma l(\gamma)$. It can be shown that this notion of distance is a geodesic distance on a Riemannian manifold. In particular, if we consider the space of parameters $\mathcal{H}$ equipped with the Fisher-Rao metric, it can be shown that this infinitesimally coincides with the KL divergence,

$$I_\mathcal{H}(\eta) = \int_\mathcal{X} \left[ \nabla_\eta \log p(x|\eta) \cdot \nabla_\eta \log p(x|\eta)^\top \right] p(x|\eta) \mathrm{d}x. \tag{4}$$

The decoder $h : \mathcal{Z} \to \mathcal{H}$ is a map where $\mathcal{H}$ is equipped with the Fisher-Rao metric. Then we can equip $\mathcal{Z}$ by pulling back the Fisher-Rao metric along $h$ given by: $M(z) = J_h^\top(z) I_\mathcal{H}(h(z)) J_h(z)$, where $J_h$ is the Jacobian of $h$.

In the sequel, we will attempt to define a similar geometry on the latent space by considering the Wasserstein distance instead of the KL divergence. We will then compute geodesics on the latent space, similar to Eq. 3, and assess if they are sensible (Figure 1).

## 3. Wasserstein Distance

Recently the Wasserstein distance, from the field of optimal transport, has been attracting increasing attention. Optimal transport theory provides a way of defining a Wasserstein metric tensor, which gives the space of probability distributions an infinite-dimensional Riemannian differential structure (Ambrosio and Gigli, 2013; Li, 2022) and it can be shown that the Wasserstein distance can be seen as the geodesic distance on this manifold. An attractive property of the Wasserstein distance is that it reflects the underlying ground metric on the sample space. This gives us reason to think that pulling back the Wasserstein metric to the latent space would better reflect the geometry since it takes into account both the distance between decoder distributions and the distance in the sample space.

The Wasserstein distance between multi-dimensional continuous probability distributions is generally intractable (Peyré et al., 2019). And while there are algorithms to approximate the Wasserstein distance between multi-dimensional discrete distributions, such as the Sinkhorn algorithm (Sinkhorn and Knopp, 1967), they are computationally expensive. However, a closed-form formula exists for the special case of Wasserstein-1 distance between 1-dimensional discrete distributions. In this case, the Wasserstein-1 distance is equivalent to the Earth movers distance (Levina and Bickel, 2001). For two discrete distributions, $p$ and $q$ of length $N$ the distance is

$$W_1(p, q) = \sum_{i=1}^{N} |\varphi_i|, \text{ where } \varphi_i = \sum_{j=1}^{i} (p_j - q_j). \quad (5)$$

Furthermore, it can be shown that for product measures, we have

$$W_1^2(\otimes_{i=1}^n \mu_i, \otimes_{i=1}^n \nu_i) = \sum_{i=1}^{n} W_1^2(\mu_i, \nu_i). \quad (6)$$

Jointly, Eqs. 5 and 6 allow us to compute and backpropagate through the Wasserstein distance in some specific setting. This in turn allows us to compute geodesics in the latent space under the pull-back Wasserstein metric.

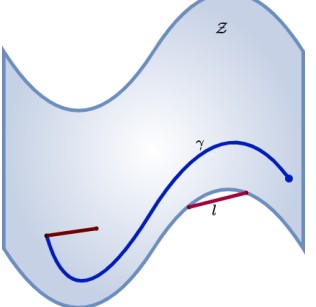

Figure 1: The intrinsic geodesic $\gamma$ and the Euclidean shortest path $l$.

We specifically consider a VAE with a decoder that outputs a discretized logistic distribution to generate images, similarly to Salimans et al. (2017) and Maaløe et al. (2019). As is common in VAEs, we consider data dimensions to be conditionally independent, i.e. $p_\theta(x|z) = \prod_{i=1}^{D} p_\theta(x_i|z)$. Each pixel is modeled as a discrete random variable taking values in one of 256 bins. Given two images, the Wasserstein distance between corresponding pixels can be computed using Eq. 5, and the Wasserstein distance between the two images can be computed using Eq. 6. Then we can use an approximation of the length of a curve as in Eq. 3 to compute the shortest path (geodesic) between two

points,

$$l(\gamma) = \lim_{N \to \infty} \sum_{n=1}^{N-1} W_1(p(.|\gamma(n/N)), p(.|\gamma((n+1)/N))). \tag{7}$$

We parametrize curves $\gamma$ as cubic splines and minimize curve energy using gradient-based optimization. This is realized with the `StochMan` library (Detlefsen et al., 2021).

## 4. Experiments

We train a VAE with a discretized logistic decoder on the dataset and compute geodesics on the latent space equipped with the pull-back of the Wasserstein metric.

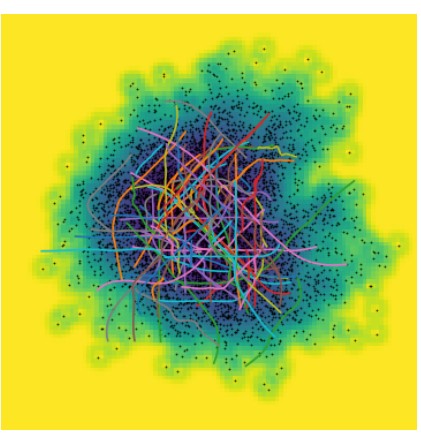

We train these VAEs on a subset of the MNIST dataset composed of only the digits with some specific label and $\mathcal{Z} \subset \mathbb{R}^2$. Figure 2 shows the geodesics on the latent space learned by VAE computed using Eq. 7. We see the geodesics lie inside the latent space.

We also train the VAE on the entire MNIST dataset with $\mathcal{Z} \subset \mathbb{R}^{30}$. We compute a geodesic on this latent space and decode various points lying on this geodesic and we compare it with linear interpolation in the space of images. As illustrated in Figure 3, the images on the

Figure 2: Latent Wasserstein geodesics.

geodesic are plausible members of the dataset, whereas the images on the linear interpolation don't always lie on the dataset(See Appendix Section A for more examples).

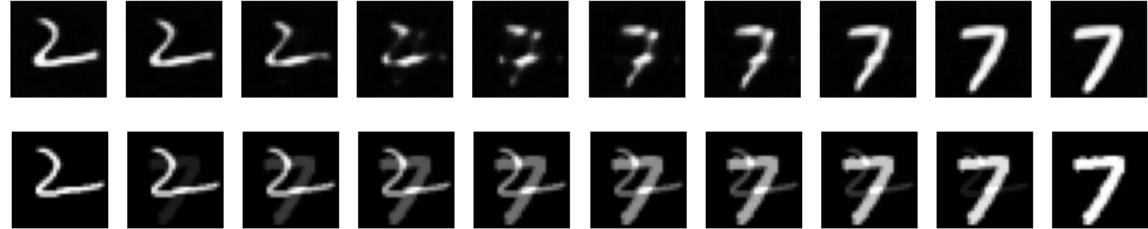

Figure 3: The top row shows images that lie on the geodesic connecting two examples from MNIST and the bottom row shows images lying on the linear interpolant connecting them.

## 5. Discussion

We have presented an efficient algorithm for computing latent space geodesics under the pull-back Wasserstein metric associated with discrete decoders. This both allows for having a deterministic metric in a stochastic model and for incorporating the observation space metric, thereby getting the best parts of existing geometries (Arvanitidis et al., 2018; Arvanitidis et al., 2022). The presented work is, however, early with several aspects still missing investigation. Previous studies have, both theoretically and empirically, demonstrated that the uncertainty of the decoder plays a crucial role akin to the topology of the learned

manifold (Hauberg, 2018; Detlefsen et al., 2022). We did not investigate this aspect here. Furthermore, in the present study, we have only investigated how to compute the geodesic connecting two latent points. While this is important, many other geometric tools are currently missing. Perhaps most elementary, we have not provided explicit access to the latent space metric. These, and more questions, will be investigated in future work.

**Acknowledgments.** This work was funded in part by the Novo Nordisk Foundation through the Center for Basic Machine Learning Research in Life Science (NNF20OC0062606). It also received funding from the European Research Council (ERC) under the European Union's Horizon 2020 research, innovation programme (757360). SH was supported in part by research grants (15334, 42062) from VILLUM FONDEN. The authors also acknowledge the support of the Pioneer Centre for AI, DNRF grant number P1.

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

## Appendix A. Comparison of geodesics and linear interpolants

We present various examples of comparisons between images lying on a geodesic and the image lying on the linear interpolant between two images.

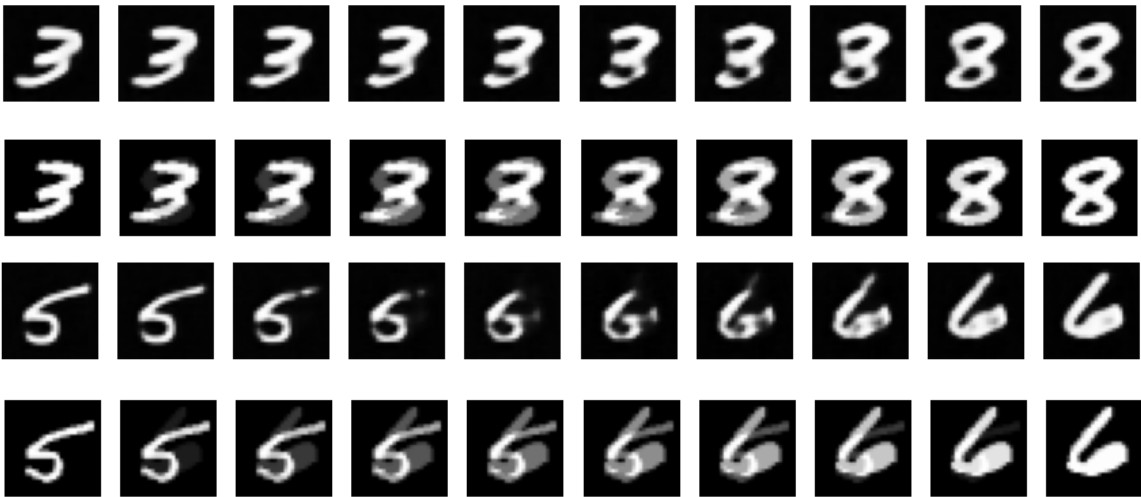

Extended Abstract Track

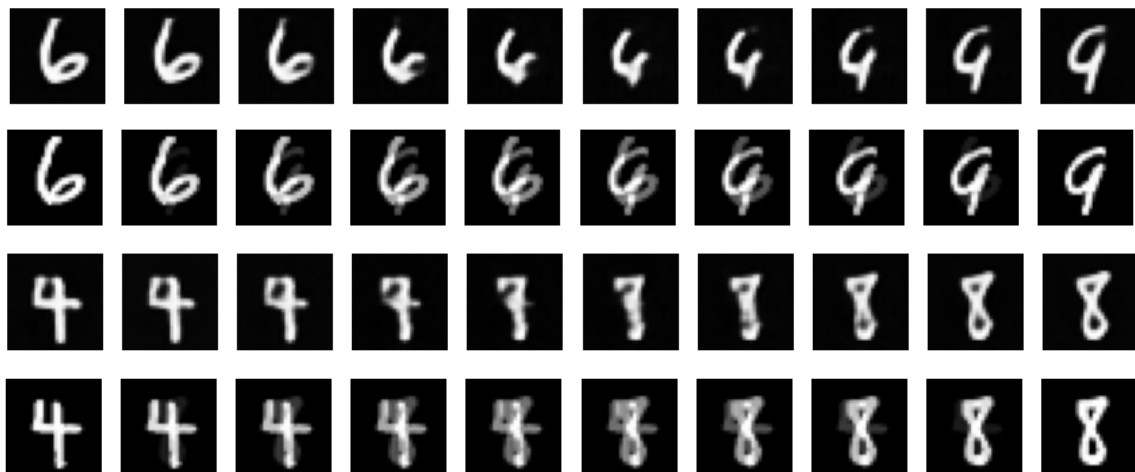

Figure 4: In these figures the top row shows the images that lie on the geodesic connecting two images of MNIST and the bottom row are the images that lie on the linear interpolant connecting the two images

