# OpenReview forum: "Optimal Latent Transport"
_NeurIPS.cc/2022/Workshop/NeurReps — NeurReps 2022 Poster_

### Official Review · Reviewer_Nub4 · 2022-10-13
**Review of Optimal Latent Transport**

**Confidence:** 3
**Soundness:** 3
**Presentation:** 3
**Contribution:** 3
**Overall Rating:** 5

**Summary:**

The authors investigate an extension of VAEs, beyond the typical Gaussian distributed latents. They use a "decoder that outputs a discretized logistic distribution to generate images", and then compute the Wasserstein distance between distributions using a closed form (Eqs. 5,6)

The authors explain how to back prop through the Wasserstein distance and how to parametrize geodesic curves between images using Eq 7.

The authors present results interpolating between two images via the geodesic, versus a pixel based linear interpolation of the images.

**Questions:**

VAEs are also being used in various scientific applications, beyond every day image data (e.g. astronomy, microscopy and spectroscopy in chemistry and materials sciences). Will your approach be applicable in these areas, and are you interested in them?

What is the connection between the ELBO loss and the computed geodesics on the latent space? Do you train first, and then afterwards explore the latent space?

At the end of page 2 you write "In the sequel,", do you mean in follow up work and not in this paper?

It's not clear how much from the section "2. Background" was used for the results (Fig 2,3) in this paper.

In Figure 1, is the red line meant to go underneath the manifold and that's why it is absent in the middle region? Perhaps put it faded or dotted...

Figure 2 is instructive and connects nicely with Fig 1. Perhaps you could match up their colours somehow.

**Limitations:**

Did you compute the linear interpolations of the images with no reference to the Gaussian-based-VAE latent space? I think it would make more sense to draw samples along a path between their latents. This is what I've commonly seen when people are visualizing a latent space with these type of VAEs. Of course it would involve implementing such a Gaussian-based-VAE... This is the reason I put an overall rating of 5 vs something higher.

**Recommended Decision:**

3: Accept

**Relevance:**

4: Highly relevant

**Strengths And Weaknesses:**

You present VAEs well, and point to literature that is extending them beyond Gaussian distributions. However you don't speak to the technical / implementation aspect of why some distributions are suitable or not, in terms of the reparametrization trick.

I did not understand your motivations explained in the first paragraph of "3. Wasserstein Distance". Perhaps you'd like to spell it out a bit more for the reader - including some additional text before the sentence "This gives us reason to think...". Perhaps you could include some specific page numbers in the references given, or additional ones, to go into more detail.

You do a good job mentioning your next steps and where you want to take this.

More results would be good, beyond interpolating images. Perhaps that go into computational expense, edge cases.

**Submission Track:**

Extended Abstract (4 Page)

---

### Official Review · Reviewer_8oaS · 2022-10-15
**Review of Optimal Latent Transport**

**Confidence:** 4
**Soundness:** 2
**Presentation:** 4
**Contribution:** 2
**Overall Rating:** 5

**Summary:**

This manuscript investigates a novel method to interpolate data in the latent space of generative models. This works by computing the pullback metric of the 1D wasserstein distance in the input space, and use that as a metric for computing geodesic trajectories in the latent space to interpolate data via cubic splines. This differs from previous works which assume a different choice of pullbakc metricw, e.g Euclidean or Fisher-Rao


**Questions:**

Which are the advantages/ properties  of choosing the wasserstein metric as a the pullback metric to compute geodesics in the alten space, compared to other choices?



**Limitations:**

Most limitations were correctly discussed by the authors in the paper.
In addition, in its current state the manuscript misses a direct comparison with approaches based on pulling back the fisher rao metric, or the euclidean one both  from a theoretical or experimental perspective.. More generally it is clear which are the advantages of pulling back the wasserstein metric compared to other choice of metric in the latent space. I think that some discussion/ theory / experimental evidence should be added to the manuscript in this regard.


**Recommended Decision:**

2: Borderline

**Relevance:**

3: Solid fit

**Strengths And Weaknesses:**

*Originality*

The work is an original investigation of an alternative metric to compute geodesics in the latent space of generative models, which builds on previous methods which rely on a pullback metric on the input space.

*Quality*

To the best of my judgement the method proposed is technically sound. Concerning experimental evaluation, results are limited to qualitative analysis on some samples.
In addition, is pretty difficult to evaluate which properties are guaranteed/ which advantages by choosing the wasserstein distance as the pullback metric

*Clarity*

The paper is clear and well written.

*Significance*

The approach is promising as an alternative approach to compute geodesics in the latent space, preserving the metric struvìcture of 1d prob distribution in the input space, therefore worth investigating . However in its current stage is still difficult as a reader to understand which advantages/ disadvantages come from the particular choice both from a theoretical or experimental perspective


**Submission Track:**

Extended Abstract (4 Page)

---

### Official Review · Reviewer_ZVSu · 2022-10-15
**Review of Optimal Latent Transport**

**Confidence:** 4
**Soundness:** 3
**Presentation:** 3
**Contribution:** 2
**Overall Rating:** 4

**Summary:**

The authors propose a method to compute the geodesics  the latent space of VAE's associated with the pull back metric of the 1-Wasserstein metric on the decoder distributions. This approach relates to previous work from Arvantidis et al. (2022), where the Fisher Rao metric was pulled back. To implement the method, the authors also use classical results to for the Wasserstein 1-distance and consider the case of a decoder that outputs discretized logistic distributions.

The authors visualize the results on the MNIST dataset on a latent space of dim 2 by showing the geodesics obtained in the latent space, and images from the geodesic trajectories on a latent space of dim 30, that differ from linear interpolation.

**Questions:**

Equation 5 has typos (p_j and q_j)

I would tone down the claim that "algorithms to approximate the Wasserstein distance (...) are still too computationally expensive to be practical". In particular, see Solomon, et al. (2015) doi.org/10.1145/2766963 , who use a Heat kernel approximation in the Sinkhorn algorithm. This was also recently implemented with GPU to quickly interpolate large 3D images with the Wasserstein metric (Ecoffet et al 2021 10.3934/math.2022059).

In dimension 2, why does the fact that geodesics lie inside the latent space matter? For example, could we imagine that the mass between two distinct images could be transported at lower cost without representing other images that are in the data space? Would it then yield some geodesics that do not belong to the distribution of images in the latent space? For two given images, could the authors also compare images decoded from a straight line in the latent space, and compare with images generated from the geodesics to see if the interpolants look more natural for example?

Results from dimension 30 seem to reflect what one would expect, i.e. linear interpolation blending the images (and teleporting mass), in contrast with the Wasserstein interpolants that continuously displace mass. It would also be interesting to compare these with the Fisher Rao metric.

Technical details about the implementation and hyper parameters and training are also missing and make it hard to assess the reproducibility and technical quality of the implementation.



**Limitations:**

The authors discuss the limitations of their approach in the Discussion. Maybe the authors could also consider that using the Wasserstein distance for distributions over the 2D space instead of a product measure of 1D distributions, would be more relevant in order to reflect the geometry of the image space.

**Recommended Decision:**

2: Borderline

**Relevance:**

3: Solid fit

**Strengths And Weaknesses:**

The approach is original to my knowledge, and the idea of using the Wasserstein metric to interpret the latent space is sound. The results are still preliminary but even in the Extended Abstract, I find them quite limited with a lack of interpretation and comments from the figures, or claims that are questionable (see further comments below). Perhaps the authors could have included some technical details as Appendix to provide more results, given the limited length of the submission, or focus on investigating in more details the results obtained in dim 30. In its current state of the paper, I find the paper potentially interesting but also needing more work to establish some good preliminary results.

**Submission Track:**

Extended Abstract (4 Page)

---

### Decision · Program_Chairs · 2022-10-21

Accept (Poster)